# Heteronormative Representations of the Family and Parenting in Public Policies: Implications for LGBTIQ+ Families

Rodolfo Morrison *, Sebastián Gallardo and Francisca Parra Fuster

Departamento de Terapia Ocupacional y Ciencia de la Ocupación, Facultad de Medicina, Universidad de Chile, Santiago 8320000, Chile
* Correspondence: rodolfo.morrison@uchile.cl

**Abstract:** This research analyzes the discourse of the Chilean State Program: "Chile Crece Contigo", a program aimed at supporting the upbringing of children. We wonder about representation in the discourses of family and parenting, especially regarding LGBTIQ+ families. To do this, we compiled the materials available on the program website, which are particularly directed at the sphere of family and parenting. We carry out a documentary analysis, based on an approach to the post-structuralist analysis of public policy to identify how and what is the representation of the problem that public policies seek to solve. The results show only heteronormative perspectives to understand families, where sexual diversity within its constitution is almost invisible. At the same time, the exercise of parenting is represented as a materialization of sexist practices that reproduce stereotypes roles. Thus, this reinforces the idea that heteronormativity establishes an articulating axis of public policy that regulates the life of the subjects, maintaining differentiating parameters regarding the consideration of an expected behavior in society based on sex/gender. It is concluded that gender stereotypes supported by heteronormative models cause the marginalization of a significant percentage of families that do not fit into the imaginary of public policies under these heteronormative models, such as LGBTIQ+ families.

**Keywords:** Chile; public policy; heteronormativity; occupational science

## 1. Introduction

In Chile, the fight to win the rights of LGBTIQ+ people has sought to break the hegemony that unites "religious and political sectors in defense of a [heterosexist] sexual order based exclusively on heterosexual marriage for procreative purposes" (Faúndes et al. 2012, p. 28). In addition, it is moving away from the conception of families in which there is only the father and mother, with their biological children (Arias 2004).

The way in which the family has been understood from a heteronormative perspective has been disputed. Heterosexuality is understood as something "natural" and the gender stereotypes are derived from this understanding. An example of this is the criticism of traditional gender roles represented as: men as the productive function (produce, bring livelihood to the home), while women have a reproductive function (work at home and raise children) (Fuentes-Barahona et al. 2022; Galaz et al. 2021; Mondaca et al. 2022; Morrison et al. 2020).

In a certain way, the heteronormative model is still valid in various discourses, despite the existence of various forms of family (Abad et al. 2013; Galaz et al. 2018). This vision of what a family is and the specific functions of each member can influence people's beliefs and relationships, affecting the way through which they decide to create/be a family, that is, the parenting of families outside of this heteronormative margin can be restricted (Gómez-Antilef et al. 2020).

On the other hand, as a central role within the family, parenting is influenced and restricted by various factors, from gender mandates, providing a model of how to be a

mother or father, to the state's vision of what a family is (Galaz et al. 2018). Parenting can be expressed in different ways: referring to people who beget, conceive, give birth, and raise children, regardless of sex/gender, based on shared responsibility, including children in various affective constellations, among other family arrangements (Alday-Mondaca and Lay-Lisboa 2021b; Verea 2005).

There is a precedent that discriminatory practices have been reproduced in Chile based on heterocisnormative conceptions of couple relationships/parenting (Barrientos 2015, 2016; Díaz and de Prada 2021; Morrison et al. 2020; Morrison 2022). Therefore, the question that guides this research is what are the representations of LGTBIQ+ families in Chilean public policy, especially in the state program "Chile Crece Contigo" ["Chile develops with you"]?

This state program is a pioneer in the country and its objective is to establish itself as a comprehensive child protection subsystem. It seeks to "accompany, protect, and support all children and their families, through an integrated system of universal social interventions and other differentiated benefits for children in vulnerable situations" (Chile Atiende 2022).

Access to this program begins with the first pregnancy control in a public health center and is carried out by the midwife with the delivery of the pregnancy and a birth guide. Likewise, boys and girls can access their first healthy check-up carried out by the nurse. This accompaniment extends to the first cycle of basic education (Chile Atiende 2022). This is a far-reaching program from the Chilean state, so we assess how all families that can access this program are represented. In particular, we are interested in identifying the ways in which families are represented, especially LGBTIQ+ families and parenting.

## 2. Family, Parenting, and LGBTIQ+ People

The family can be understood as the relationships that arise to ensure the fundamental forms of human dependency, such as birth, child rearing, emotional and support bonds, generational ties, suffering from a disease, and death (Butler 2010). In some families there is the exercise of parenting. Parenting can be understood as the term that acquires the ability to point out those characteristics attributable to adult and care figures, without specific distinctions (Waissbluth 2016). Thus, the family can be the context in which parenting will have its niche of action.

Parenting, understood as an occupation, or significant activity, considers the care of boys and girls as a highly significant and complex element of human existence, since it is considered an occupation in which more than one person participates and implies the acquisition of new roles (Domínguez et al. 2018). Parenting is exercised by people who engender, conceive, give birth, and/or raise children, without any distinction of sex or gender, taking into account that it is a shared responsibility, which includes children within this dynamic (Verea 2005). Likewise, parenting takes into consideration the importance of both social and historical influences (Klug and Bonsall 2014). Therefore, it is extremely relevant since it implies planning and organization, which leads to transformations both in the meaning of the occupation, as well as in the new roles that must be adopted and in the challenges that it brings to occupational participation (Lim et al. 2022; Domínguez et al. 2018; Sethi 2020).

The exercise of parenting is a process that is influenced by sociohistorical events (Sethi 2020). In this context, institutionality delimits and regulates the recognition of parenting (Cilliers 2021). In Chile, the research that has accounted for the exercise of parenting has been quite diverse. Some show how couples carry out actions inside and outside the institutional framework to become parents (Alday-Mondaca and Lay-Lisboa 2021a; Morrison 2022); another research has shown how socioeconomic differences are determinant in becoming parents (Herrera et al. 2018).

In this sense, in Chile, the approval of equal marriage recently occurred, where families of people from the LGBTIQ+ community were legally recognized (Ley No. 21.400 2021). In this regard, it is important to ask how these families are represented in public state policy.

Is it possible that these families will continue to become invisible because of the state? How are the set of state institutions, procedures, and programs related to these families?

The representations of the state, materialized in public policy, can say how families can be understood at different levels. For example, in the visual illustrations of family-oriented brochures, in the written texts of parenting ministry programs, etc. These representations can influence the processes of subjectivation of LGBTIQ+ families and, therefore, the exercise of parenting.

This situation produces subjects granting them certain characteristics that will determine, recursively, the performance of parenting. This way, the policies perpetuate a fictitious centrality and an externalizing marginality that in their daily life configure subjectivities categorized as 'different'. 'others', or 'diverse', as opposed to a social group that maintains and reaffirms, in turn, a privileged position, under categories within the "normal" (Cánovas and Molina 2015).

However, the progress in matters of rights and legitimization of same-sex couples is recognized, but the literature has indicated that this helps to solve some problems related to representation derived from the upbringing raised by two mothers or two fathers (Crouch et al. 2014; Hermosa-Bosano et al. 2022; Tombolato et al. 2018). In addition, it is important to comment that LGBTIQ+ families have a counter-hegemonic conception compared to the Western idea of family, founded by the Judeo–Christian tradition, which does not mention other ways of performing parenting (Libson 2012; Maqueda and Salinas-Quiroz 2020). It is under the previous premises that other questions arise, such as: How are the representations expressed by the Chilean state in relation to family and parenting and, especially, what refers to LGBTIQ+ families? To approach this question, we use the state program "Chile Crece Contigo", which is focused specifically on this issue.

In the field of social sciences, answering this question could contribute to highlighting the stereotypes and understandings of family and upbringing that exist in public and state programs. What could generate different international dialogues in this matter.

## 3. Methodological Framework

This research seeks to identify discourses on family and parenting, with an emphasis on LGBTIQ+ families. For this, a qualitative design of a documentary case study was used, which allows us to approach and explain the processes by which different situations connect and live in the world and are expressed through the materiality of the documents (Tancara 1993). Furthermore, this design causes it to be possible to understand, describe, analyze, or interpret these interactions in specific sociohistorical contexts (Flick 2015). The documentary case study seeks, through the collection and analysis of files and other documents, to process and characterize the information contained in the documents and the systematic, coherent, argued, and analyzed presentation of new information in a scientific document (Martinez 2011).

Purposeful selective sampling was used (Martinez 2011), where the sample unit corresponds to materials obtained from the website of the "Chile Crece Contigo" program in "Materiales" [Materials] section, subsection "Material para Familias y cuidadores" [Materials for Families and Caregivers]. From the 96 materials available on the website, 18 were selected and analyzed under the following criteria (see Table 1):

Inclusion criteria:

- Those related to parenting.
- Those containing terms such as "paternity", "maternity", and/or "nurture".

Exclusion criteria:

- Materials that address issues related to childbirth.
- Materials that address breastfeeding and the feeding of children.
- Materials that address issues of diversity between cultures.

**Table 1.** Material analyzed.

| No. | Type of Material | Material Name | Material Creation Date | Recovered from |
|---|---|---|---|---|
| 1 | Poster | 7 keys to active parenting | 2016 | https://www.crececontigo.gob.cl/wp-content/uploads/2016/03/7-claves-para-una-paternidad-activa.pdf |
| 2 | Triptych | Maternity and paternity Labor Rights Primer | 2016 | https://www.crececontigo.gob.cl/wp-content/uploads/2016/03/Derechos_laborales-de-materinidad-y-paternidad-V2020.pdf |
| 3 | Poster | 5 keys to support the family of a newborn | 2016 | https://www.crececontigo.gob.cl/wp-content/uploads/2016/03/Afiche-recien-nacido-50x70-cm-CS6.compressed.pdf |
| 4 | Guidebook | Guide to Active Paternity in Education | 2017 | https://www.crececontigo.gob.cl/wp-content/uploads/2017/08/Guia-Paternidad-Activa-en-Educacion-final.pdf |
| 5 | Guidebook | I am here! | 2017 | https://www.crececontigo.gob.cl/wp-content/uploads/2017/09/cartilla-ya-estoy-aqui.pdf |
| 6 | Opinion column | Looking at same-sex parented family from parenting | 2017 | https://www.crececontigo.gob.cl/columna/mirando-la-homoparentalidad-y-la-lesbomaternidad-desde-la-crianza/ |
| 7 | Book | Nobody is perfect. Parenting skills workshop for mothers, fathers, and caregivers of children from 0 to 5 years old. Part C: Fathers, Mothers, Caregivers(s) | 2018 | https://www.crececontigo.gob.cl/wp-content/uploads/2020/04/Nep_Padres-Madres-y-Cuidadores-V2019.pdf |
| 8 | Book | Nobody is perfect. Parenting skills workshop for mothers, fathers, and caregivers of children from 0 to 5 years old. Part D: Mental Development | 2018 | https://www.crececontigo.gob.cl/wp-content/uploads/2020/04/Nep_Desarrollo-mental-V2019.pdf |
| 9 | Book | Nobody is perfect. Parenting skills workshop for mothers, fathers, and caregivers of children from 0 to 5 years old. Part E: Physical Development | 2018 | https://www.crececontigo.gob.cl/wp-content/uploads/2020/04/Nep_Desarrollo-fisico-V2019.pdf |
| 10 | Book | Nobody is perfect. Parenting skills workshop for mothers, fathers, and caregivers of children from 0 to 5 years old. Part A: Behavior | 2018 | https://www.crececontigo.gob.cl/wp-content/uploads/2020/04/Nep_Comportamiento-V2019.pdf |
| 11 | Diptych | Behavior Problems in Boys and Girls from 5 to 9 years old | 2018 | https://www.crececontigo.gob.cl/?s=Problemas+de+Conducta |
| 12 | Opinion column | Same-sex parenting and upbringing: mother and father as a function | 2018 | https://www.crececontigo.gob.cl/columna/homoparentalidad-y-crianza-madre-y-padre-como-una-funcion/ |
| 13 | Primer | 10 things your child needs | 2019 | https://www.crececontigo.gob.cl/wp-content/uploads/2019/04/10-cosas-que-tu-hijo-necesita.pdf |

Table 1. *Cont.*

| No. | Type of Material | Material Name | Material Creation Date | Recovered from |
|---|---|---|---|---|
| 14 | Book | Nobody is perfect. Parenting workshop for families of boys and girls from 5 to 9 years old. Conduct | 2020 | https://www.crececontigo.gob.cl/wp-content/uploads/2020/02/Manual-NEP-Conducta.pdf |
| 15 | Primer | Collection: Respectful parenting primers. Active paternity | 2021 | https://www.crececontigo.gob.cl/wp-content/uploads/2022/04/2-Paternidad-activa_2021.pdf |
| 16 | Primer | Important newborn care | 2021 | https://www.crececontigo.gob.cl/wp-content/uploads/2021/04/Cuidados-Importantes-del-Recien-Nacido_2021.pdf |
| 17 | Primer | Collection: Respectful parenting primers. Respectful parenting | 2021 | https://www.crececontigo.gob.cl/wp-content/uploads/2022/04/Cartillas-de-Crianza-Respetuosa_2021.pdf |
| 18 | Book | Discovering together the development and stimulation of your son or daughter in their first two years of life | 2021 | https://www.crececontigo.gob.cl/wp-content/uploads/2022/04/Descubriendo-Juntos-2021.pdf |
| 19 | Primer | Collection: Respectful parenting primers. Stimulation | 2022 | https://www.crececontigo.gob.cl/wp-content/uploads/2022/04/15-Estimulacion_2021.pdf |
| 20 | Primer | Collection: Respectful parenting primers. Attachment | 2022 | https://www.crececontigo.gob.cl/wp-content/uploads/2022/04/4-Apego_2021.pdf |

Additionally, two opinion columns were added, since it was the only input of the program that made explicit the role of the same-sex parented family (materials No. 6 and 12). For the analysis process, an approach to critical discourse analysis was used, which corresponds to an analysis technique that consists of addressing the use of language as a data source to identify dialects and styles, recognize social identities, and reconstruct discursive representations and intertextual games (Van-Dijk 2017). In addition, as an analytical framework, an approach to the post-structuralist analysis of public policy was used (Bacchi 2009). From this analysis, it is assumed that the problems do not exist a priori but are represented by the construction of public policy. This perspective challenges the idea that governments respond to preexisting problems and argues instead that they are actively creating or producing those "problems" (Bacchi 2012). This does not mean that the problems or experiences that a policy addresses are not real but, instead, it understands these conditions to be represented as a social problem.

## 4. Results

From the 20 materials analyzed, we identified several common issues. The first is the permanent reproduction of a family stereotype. The permanent replica of an ideal family was identified as the roles the people who constitute a family nucleus should have. For example, there is a workshop called "Nobody is perfect" (material no. 8), which is divided into several sections, one of them directed to fathers, mothers, and caregivers, which states: "sons and daughters should have an approach with adults of the same sex", suggesting that for parenting a specific sex/gender "model" is required in girls or boys.

In several of the materials, heteronormative patterns are reinforced (among which the materials from No. 2 to 10 and No. 14 stand out). These materials express content framed in gender roles that do not consider LGBTIQ+ families. Particularly, when referring to the need for members of different sexes in parenting processes and in the division of tasks by

gender/sex. Regarding paternity, for example, Material No. 4 provides a specific guide to the tasks, reinforcing what should or should not be done while raising children: "A present, committed and affectionate father positively influences the learning, development and well-being of their daughters and sons in various areas". In the case of motherhood, in most of the materials on behavior management in children, the image of a mother appears. Additionally, when representations of the father appear, he is usually in outdoor spaces, e.g., riding a bicycle with his son (material No. 10, p. 22).

The construction of paternity is interesting. There are a series of documents oriented toward active paternity, where a certain distance is assumed within the parenting process. For example, in material No. 1, the following indications are provided: "Build a bond with your child from the news of the pregnancy" and "Be the protagonist of his daily care: calm his crying, move him, change his clothes, bathe him, prepare his food, feed him." A priori, this representation assigns the man a place outside the parenting process, where he must try to be involved.

Both in issues such as attachment, parenting, and care for newborns, it is assumed that the main caregivers are a family constituted of a biological mother and father (highlighting materials No. 4, 5, 18, and 20). The man is considered someone who is not active in parental occupations, as a companion and/or facilitator, showing that men and women complement each other in parenting, leaving out single-parent families and LGTBIQ+ in their recommendations. Even when referring to blended families, where several LGBTIQ+ families could be considered (Iguales 2020), there is no such mention of this whatsoever.

In the Primer on labor rights (material No. 2), most of the rights related to the care of the newborn are associated with the mother and are extended to the father only if the mother wishes so, if he is the legal guardian of the child or if the mother dies. In addition, the father is framed under a productive role (worker), which can reinforce the idea of gender roles. Likewise, by establishing gendered roles, same-gender couples are left without a clear notion of their work regarding their labor rights in the gestation process. "In Chile there are inalienable labor rights for pregnant women who have an employment relationship. These rights extend to the worker (father) in certain cases and can be permits, subsidies, and privileges" (material No. 2, p. 1).

On the other hand, although in practically all materials, representations of the family such as a father, mother, and child abound; in chapter nine of material No. 7, the theme "single-parent families" is found. In this section, it is the only moment in which there is an explicit reference to other types of families, however, it continues to be carried out from a heteronormative conception and from a deprivation perspective, where these families would not reach the ideal: "Although the parents and mothers do not live together, they still have a mom and dad like other boys and girls" and "Comfort your sons or daughters and give them all the time and support they need" (p. 24).

On the other hand, in the different materials that were reviewed, there is an assumption in which families with good support networks and ideal social skills are visualized to request support when problems arise. Particularly in the field of mental health, it is advisable to be linked to people who provide certain socio-emotional support. However, it omits and causes the families that do not have consolidated networks to be invisible and, therefore, the recommendations are not delivered in a situated manner. There is no consideration that there are families that do not have the time or necessary means to generate new ties. Therefore, the recommendations for a number of families do not present any feasible solution, but, rather, it could be the cause of a certain state of suffocation, by not opting for these conditions immediately.

Finally, regarding the only two materials found that talk about same-sex parented families, one of them explicitly states that: "We know that when talking about father and mother we are talking about functions" (material 12, p. 1), thus implying that there would be roles of "feminine and masculine" predesigned in families. While the other offers a demystification against various prejudices about the upbringing of children by same-sex couples. This is the only material that refers to studies that show that there

are no differences in terms of the impact on children, in the upbringing of same-sex or different-sex couples:

> *"[B]y considering the aspects of having a loving and well-treated space for boys and girls, we can get involved in such a way that as a society we can provide the space for homosexual and lesbian parent families to raise and accompany the development of girls and boys, having a space visible and secure within the society"*. (material 6, paragraph 10)

## 5. Discussion

From a critical discourse analysis, we can identify that one of the beliefs present in the analyzed material about the need for a heteronormative family model, that is, father and mother, is supported by the assumption that modeling in the socialization process for gender is related to good upbringing (Leaper 2014). Clearly, this is problematic in the case of LGBTIQ+ families, since these family models do not have figures of different sexes. In this regard, the parental skills of mothers and fathers can be questioned when describing sex/gender as necessary within the socialization processes (Herrera et al. 2018; Morrison et al. 2020). These assumptions are rooted in a heteronormative system that is not quested (Barker 2014). This system reproduces and disseminates, through various mechanisms, behavior patterns that boast of being normal (Bell 2009), without considering family diversity (Abad et al. 2013), or what multiple studies have indicated about raising children (Agustín 2014; Silva et al. 2022; Tombolato et al. 2018).

This way, social institutions perpetuate the hegemonic gender logic, causing it to be very difficult to generate transformations that reduce the oppression faced by subjects who are judged by heteronormative logics (Galaz et al. 2016). For example, heteronorms, in combination with a strong idea of maturation in the aging process, puts a lot of pressure on queer children and adolescents. In a culture dominated by the discourse of "family values", the outlook is bleak for any hope that the institutions linked to the family and the state that deal with child rearing would become less oppressive (Warner 1991). The same occurs with the invisibility of LGBTQ+ families, the construction of a series of public policies that do not consider them, and the construction of their subjectivation representing an idea of otherness and remoteness (Hollekim et al. 2012).

As has been pointed out internationally (Rubin 1984), the construction and organization of our society in terms of gender/sex inevitably affects the different ways of functioning in society. For example, societies that are more respectful of LGBTQ+ people provide indicators of more favorable mental health for their population (Ventriglio et al. 2022). This way, public policies are not innocuous in the representation they have of the sex and gender of people. A non-oppressive gender can only emerge through a radical change in the understanding of sexuality and, for this, it is essential to threaten the heteronormative expressions that house, in different institutions, homophobic and heterosexist expressions (Warner 1991) and are expressed/materialized in public policy.

Parenting is influenced by different factors, where the representations governed by the heteronorm have repercussions on how this occupation is exercised (Alday-Mondaca and Lay-Lisboa 2021a; Herrera et al. 2018). On the other hand, in the analyzed material, the representation of single-parent families is presented as an alternative that would never be chosen. Single parenting is represented as a consequence of a problem that occurred with the couple, such as separation or divorce. The indications of the program lie in consoling the children and in the explanation that it continues to approach, in a certain way, heteronormativity.

In this regard, in the process of understanding parenting, it is necessary to integrate the relationship with the sociohistorical context that influences the actions or occupations that its performance entails. For example, Sethis' study enabled visualizing that the responses of mothers in various settings are better understood as transactional relationships between the mother's historical context, the challenges of the present, and the projection of a successful future for her children. Thus, occupations in parenting can be understood in a more

complex way from "temporary permeability" and not instantaneous (Sethi 2021). This implies a more complex understanding of parenting and is not only focused on the ways in which the family is constituted.

Another investigation (Sethi 2020) refers to the need for a broader approach that allows for analyzing the social influences on the occupations of parents. Specifically, the findings of this study show how mothers represent a series of interconnected roles and propose how studying parenting as a relational role is broader than just thinking about it as an occupation or co-occupation, which is the way in which it is represented in the analyzed material.

As another relevant aspect, the limited representations of masculine parenting have permeated the perceptions of the capacities of homosexual men. This is expressed in a low assessment of the performance of their role as parents (Araldi and Serralta 2019; Herrera et al. 2018; Maqueda 2018; Morrison et al. 2020). In this sense, the material analyzed could produce a paradoxical effect, which is a situation described in various ways of representing problems in public policies (Bacchi 2009), since it would try to improve a problem but at the same time reinforce it.

Regarding this point, various international authors have understood parenting as a skill and not as a natural instinct (Lim et al. 2022). What would allow its development changing the notion of instinct for ability or from a relational logic stresses the heteronormative model that attributes some natural characteristics to women over men.

From a masculinity point of view, in the case of gay men, parenthood has been approached from different perspectives. Some have reported difficulties in the process, those that are linked to prejudices and limitations of public policy (Herrera et al. 2018), and others to the very internalized and limited idea of the paternity exercise, due to the lack of social representation of gay fatherhood in men (Maqueda 2018). Both issues are aspects that public policies built from inclusive approaches could help to correct.

As it has been studied in several situations, many times the attempt to solve a problem generates a paradoxical effect, which is to reinforce it (Bacchi 2009). For example, it happens when mentioning what is related to the same-sex parented family. Thus, conditions are generated that would leave LGBTIQ+ families subject to inclusion processes or in a situation of exclusion in what is understood as *Occupational Apartheid* (Apablaza 2018; Moraga 2017; Morrison et al. 2020; Núñez et al. 2019; Pollard et al. 2009). This process implies the construction of injustices toward vulnerable groups from public policies that generate people unable to perform occupations or activities that are significant to them. Which would produce an occupational injustice due to the fact that, through these sociohistorical factors and the context of public policy, it would restrict the occupational participation of these families (Kinsella and Durocher 2016).

Along with the above, the different forms of representation of the images present in the documents replicate differences between the constructions of maternity versus paternity. In the first case, motherhood is represented as part of the private life, while fatherhood is represented as something public. This is similar to a study that specifically analyzed the images in some materials (Maldonado 2020). This metaphor of the inside-feminine and outside-masculine feeds a series of sexist stereotypes about the role of parenting and about the role of each person according to their gender/sex. In this stereotype, mothers would do "serious" work while fathers would do the "fun" work (Valiquette-Tessier et al. 2019). All in all, and without departing from a heteronormative look, this material tries to carry out a more active exercise of paternity considering several antecedents that it exposes, such as the inequality in tasks within the home in heterosexual couples.

Regarding family networks, the representation of an extended support network is far from the reality of Chilean families. This phenomenon has been studied from different perspectives (Torralbo 2013). The atomization of the family is something described in the literature as a process related to the neoliberal model (García-Ruiz and Sánchez-Barea 2013). However, in these materials, the ideal of the patriarchal family prevails, which is often far from reality.

Regarding the particular materials referring to same-sex couples, material 12 is quite limited as it refers only to the functionality of masculine and feminine roles in parenting. This idea reinforces the logic of the sexual division of labor by sex/gender, where there would be "female" tasks and "male" tasks that can be performed by people of the same sex. This is an idea rooted in heteronormativity that pigeonholes LGBTIQ+ families in a patriarchal model that seeks to resist changing its structure (Maqueda and Salinas-Quiroz 2020).

On the other hand, material 6 seems to be the only one with a focus on more realistic perspectives on same-sex parenting. By relying on different studies on the upbringing of children by same-sex couples, it allows for demystifying heteronormative prejudices by providing a fairly clear guide to parenting.

Bacchi's (2009, 2012) proposals on the representation of the problem are very useful to favor the observation of who public policy is targeting. In this case, the representation of a heterosexual family with external support networks is clear. However, in addition, the assumption is established that it is the mother who has a greater responsibility, at the same time as greater skills for care, therefore, the specific materials aimed at fathers try to compensate for this idea of lower capacities. Finally, same-sex couples are not mentioned in any didactic material, only in two opinion columns, where only one reports studies on the equal conditions in which children are raised.

## 6. Conclusions

Given the questions that led to this research, a series of interesting findings were revealed regarding the vision that the documents have of families in relation to gender roles and parenting occupations. In addition, despite having little material aimed at talking about LGBTIQ+ families as such, it was possible to analyze, through silencing, the way in which these families become invisible because of Chilean public policy.

In certain results and analyses of this research, the judgments associated with the way in which parenting tasks are carried out by people linked to the male gender were pointed out in some way. This omits, without problematizing, how performing parenting in LGBTIQ+ families would be. Particularly, it is those constituted of men, therefore, that the program impacts when it reinforces the idea that men would have "less instinct" regarding care work. Thus, they are assigned this imaginary of not being competent to raise or correctly carry out home and care tasks. This contributes to the occupational restriction of parenting. In this regard, several investigations have pointed out how sexism restricts occupational opportunities for LGBTIQ+ people (Avillo et al. 2015; Cerón and Morrison 2019; Fuentes-Barahona et al. 2022; Hadden et al. 2020; Lukas et al. 2021).

The gender stereotypes supported by heteronormative models produce a marginalization of a significant percentage of families that do not fit into the imaginary of public policies under heteronormative models. In the case of the "Chile Crece Contigo" program, only *one* family model is conceived. This situation contributes to the invisibility of LGBTIQ+ families.

Finally, we consider this particular Chilean case could contribute, in the context of an international dialogue, to how heteronormativity is represented in public policy and how this affects people's lives. This is part of several lines of research that study the materiality of public policy in the lives of subjects (subjectivation) (Energici 2016; Piedrahita 2014) and from perspectives that adopt the "glocal" as a necessary exercise that implies projecting particular and local situations toward an international perspective (Auyero 2001; Roudometof 2015). In this case, without doubt, heteronormativity, as a Western construct, affects the subjects differently depending on the way of life that their contexts allow them. Analyzing these contexts in detail and comparing them qualitatively would enable thinking in a more global way about alternatives to plan the production processes of public policies at a local level, addressing critical knots, recommendations, good practices, and satisfactory experiences, especially for historically vulnerable groups.

## 7. Recommendation

It is important to continue with the problematization in relation to gender roles and how these have an impact on the exercise of parenting in same-sex parented family. Likewise, with the visibility of these, through the critical analysis of materials, articles, images, etc. that are present in our daily life and above all those that are provided by state agencies.

## 8. Study Limitations

It should be noted that this is not a critique that seeks to question the actions that the program carries out but to incorporate a perspective that stresses and questions how the representations about the subjects are carried out in the provided information that is provided that produces public policy. On the other hand, this study is limited because a small portion of public policy was analyzed and the people in charge of the processes were not interviewed. These last two aspects would be interesting to consider in future lines of research.

**Author Contributions:** Conceptualization, R.M., S.G. and F.P.F.; methodology, R.M., S.G. and F.P.F.; formal analysis, R.M., S.G. and F.P.F.; investigation, R.M., S.G. and F.P.F.; resources, R.M., S.G. and F.P.F.; writing—original draft preparation, R.M., S.G. and F.P.F.; writing—review and editing, R.M.; supervision, R.M.; project administration, R.M.; funding acquisition, R.M. All authors have read and agreed to the published version of the manuscript.

**Funding:** This research was funded by Agencia Nacional de Investigación y Desarrollo, (ANID) Chile, grant number: FONDECYT de Iniciación no. 11220183. And The APC was funded by FONDECYT de Iniciación: 11220183: "Familias LGBTIQ+ y acción política del estado chileno: el parentesco y la filiación entre relaciones de poder y resistencia"[ LGBTIQ+ Families and political action of the Chilean state: relations of power and resistance with kinship and filiation].

**Institutional Review Board Statement:** The study was conducted in accordance with the Declaration of Helsinki, and approved by the Human Subjects Research Ethics Committee [Comité de Ética de Investigación en Seres Humanos] of Medicine Faculty of University of Chile [de la Facultad de Medicina de la Universidad de Chile] (Proy. No 014-2022, acta No 004; approved on 19 April 2022).

**Informed Consent Statement:** Not applicable.

**Data Availability Statement:** All the material analyzed can be found at: https://www.crececontigo.gob.cl (accessed on 21 September 2022).

**Conflicts of Interest:** The authors declare no conflict of interest.

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
