# Peer review of "Heteronormative Representations of the Family and Parenting in Public Policies: Implications for LGBTIQ+ Families"

_socsci, doi:10.3390/socsci12020066_

Round 1

Reviewer 1 Report

Please translate the Spanish terms into English!

I think this is an interesting paper. However, I have doubts about the methodology and analytic framework. I understand that author(s), in the end, choose to apply a qualitative analysis. That is acceptable, even if it is somewhat arbitrary in some cases.  

Author Response

Thank you so much for the comments. We carry out the translation, we specify the methodology. We changed the organization of results separating it from discussion and we adapted the conclusion. In addition, we will send the writing to an English proofreading service.

Additionally:

- All modifications are made with change control.

- We introduced some bibliographical references.

- We modified the table, since we added a material that we forgot in the first version of the writing.

- We changed the term parenthood to parenting, which is more consistent with the objective of the work. (In Spanish it is the same word, that is why we get confused, “parentality”).

Reviewer 2 Report

Comments on Heteronormative representations of the family and parenting in public policies. Implications for LGBTIQ+ Families

The study is an analysis of the Chilean State Program: "Chile Crece Contigo" which supports a heteronormative upbringing of children, setting aside discourses of families and parenthood concerning LGBTIQ+ families.

Kindly consider the following comments:

1.       The manuscript lacks soundness for the following reasons:

a.       Research questions were not stated in the article.  Consider stating them clearly so the readers know what the study wants to answer.

b.       The qualitative design stated as the Methodological Framework is too broad.  I suggest that a more specific qualitative design be specified.

c.       The manuscript identified post-structuralist analysis and critical discourse analysis (CDA) as methods of analysis, none of which were not clearly shown in the Discussion. I suggest that a clear delineation of Results and Discussion be made. I also suggest that PSA and CDA be clearly shown in the Discussion section. The Results and Discussion are not clearly presented.  The section Analisis collapsed the Results and Discussion of the paper.

d.       The Conclusion should be reworked.  New discussions were introduced in the Conclusion when it should be the place for stating the outcome of the study.  No new findings should be introduced in the Conclusion.

2.       Section 1.2 looks like a Literature Review.  I suggest that it be identified as such. However, if it is indeed the Literature Review, a more robust presentation of the Review of Literature should be presented. As it is, only a few studies were cited. Literature Review may be improved by considering the most recent studies on the issue.

3.       The Table on page 4 is not adequately labeled. I suggest that proper identification of the table be made. Also, the language used in some items is not English; it would be better that a translation be included.

4.       Please check the citations and references; some entries do not adhere to APA format.

5.       The title, Heteronormative representations of the family and parenting 2 in public policies. Implications for LGBTIQ+ Families, indicates that implications for LGBTIQ+ families will be offered.  Such is not evident in the Discussion.  I suggest that Results be organized around some themes for clarity.

6.       Consider separating the Conclusion from the Recommendation.

7.       In the Abstract, it is written: As a conclusion, I propose …”  This line seems inappropriate because the Conclusion refers to the outcomes of the study; to propose is to recommend. I suggest that the Abstract be revised to make it more descriptive of what the article intends to accomplish.

8.       The article is replete with misspelled words and grammatical lapses throughout the manuscript.  Some sentences are also ambiguously stated.  For examples:

a.       “… this model …” (line 33).  Which model? You may actually mean “heteronormative mode.” Please clearly specify.

b.       “Within the family, the exercise of parenthood…” What do you actually mean by this?

c.       Lines 97-102 are vague.  Some other paragraphs are vaguely expressed as well.

There are also some words used that may not be the right words to use:  for examples:

d.       The word “harshly” (line 31) may be too subjective.

e.       “This very reduced vision …” (line 34) may again be too subjective.

9.       Consider subjecting the article to Grammarly or other grammar check devices.  Some are not even complete sentences. Some sentences were stated loosely. This article would benefit from close editing.

Author Response

 First of all we would like to thank you for reviewing our manuscript. We know that revision is hard work, especially when the writing in English is not the best. We made our efforts to translate the article, but considering the comments, we will take the English writing service offered by the publisher.

Your comments were very valuable to improve our work. Thanks for all the work that went into this.

  1. The manuscript lacks soundness for the following reasons:
  2. Research questions were not stated in the article.  Consider stating them clearly so the readers know what the study wants to answer.

We added the research question in the introduction. The question is: what are the representations of LGTBIQ+ families in Chilean public policies, especially in the state program Chile Crece Contigo?

  1. The qualitative design stated as the Methodological Framework is too broad.  I suggest that a more specific qualitative design be specified.

We describe the methodology more specifically. It was described as follows:

This research seeks to identify the discourses on family and parenthood, with an emphasis on LGBTIQ+ families. For this, a qualitative design of a documentary case study was used, which allows us to approach and explain the processes by which different situations connect and live in the world, and are expressed through the materiality of the documents (Tancara, 1993). Furthermore, this design makes it possible to understand, describe, analyze or interpret these interactions from specific sociohistorical contexts (Flick, 2015). The documentary case study seeks, through the collection and analysis of files and other documents, the processing and characterization of the information contained in the documents, in the first instance; and the systematic, coherent, argued and analyzed presentation of new information in a scientific document, in second instance (Martinez, 2011).

  1. The manuscript identified post-structuralist analysis and critical discourse analysis (CDA) as methods of analysis, none of which were not clearly shown in the Discussion. I suggest that a clear delineation of Results and Discussion be made. I also suggest that PSA and CDA be clearly shown in the Discussion section. The Results and Discussion are not clearly presented.  The section Analisiscollapsed the Results and Discussion of the paper.

We rewrote the results, separating them from the discussion. In addition, we were more explicit in the contributions of PSA and CDA in the discussion.

  1. The Conclusion should be reworked.  New discussions were introduced in the Conclusion when it should be the place for stating the outcome of the study.  No new findings should be introduced in the Conclusion.

We reorganized the conclusion by removing the new discussions and rearranging the paragraphs to make them more relevant to the proposal.

  1. Section 1.2 looks like a Literature Review.  I suggest that it be identified as such. However, if it is indeed the Literature Review, a more robust presentation of the Review of Literature should be presented. As it is, only a few studies were cited. Literature Review may be improved by considering the most recent studies on the issue.

In this section we seek to provide the theoretical supports with which we approach the problem. It is not a review as such. Anyway, we integrate updated references and develop some points further.

  1. The Table on page 4 is not adequately labeled. I suggest that proper identification of the table be made. Also, the language used in some items is not English; it would be better that a translation be included.

We modify the table.

  1. Please check the citations and references; some entries do not adhere to APA format.

We check references using a reference manager.

  1. The title, Heteronormative representations of the family and parenting 2 in public policies. Implications for LGBTIQ+ Families,indicates that implications for LGBTIQ+ families will be offered.  Such is not evident in the Discussion.  I suggest that Results be organized around some themes for clarity.

By organizing the results, I hope this point has become clearer.

  1. Consider separating the Conclusion from the Recommendation.

We separated it for clarity.

  1. In the Abstract, it is written: As a conclusion, I propose …” This line seems inappropriate because the Conclusion refers to the outcomes of the study; to propose is to recommend. I suggest that the Abstract be revised to make it more descriptive of what the article intends to accomplish.

We change the abstract according to the suggestion.

  1. The article is replete with misspelled words and grammatical lapses throughout the manuscript.  Some sentences are also ambiguously stated.  For examples:

  1. “… this model …” (line 33).  Which model? You may actually mean “heteronormative mode.” Please clearly specify.
  2. “Within the family, the exercise of parenthood…” What do you actually mean by this?
  3. Lines 97-102 are vague.  Some other paragraphs are vaguely expressed as well.

There are also some words used that may not be the right words to use:  for examples:

  1. The word “harshly” (line 31) may be too subjective.
  2.      “This very reduced vision …” (line 34) may again be too subjective.

 Thank you. We will send the writing to a review in English to resolve this type of situation.

  1. Consider subjecting the article to Grammarly or other grammar check devices.  Some are not even complete sentences. Some sentences were stated loosely. This article would benefit from close editing.

We Will do it. Thank you

--

Thank you so much for the comments. We carry out the translation, we specify the methodology. We changed the organization of results separating it from discussion and we adapted the conclusion. In addition, we will send the writing to an English proofreading service.

Additionally:

- All modifications are made with change control.

- We introduced some bibliographical references.

- We modified the table, since we added a material that we forgot in the first version of the writing.

- We changed the term parenthood to parenting, which is more consistent with the objective of the work. (In Spanish it is the same word, that is why we get confused, “parentalidad”).

Round 2

Reviewer 2 Report

1.       There are still sections where parenthood (instead of parenting) is used.

2.       In the abstract, the last sentence, still needs some editing: 

a.       As final reflections, we propose the need to include different perspectives that include a greater variety of families, beyond traditional representations. Especially within public policies.

b.       The investigations that have accounted for the exercise of parenthood, in this sense, in Chile, have been quite diverse. Where couples must develop a series of different strategies to become parents (Alday-Mondaca & Lay-Lisboa, 2021a; X 92 2022) where socioeconomic differences are usually determinant to achieve this, not necessarily in relation to institutionality (Herrera et al. , 2018).

c.       It does not argue that the problems or experiences that a policy addresses are not real, but instead calls these conditions a represented social problem.

d.       Along with the above, the images show representations that replicate stereotypes in 387 parenting in the logic of the private-internal, regarding motherhood, versus the public-388 external, where paternity is linked. Which is similar to a study that specifically analyzed the images in some materials (Maldonado, 2020).

3.       I suggest that Table 1 be simplified; in fact, its inclusion may be unnecessary.  

4.       Some lines are still written vaguely, for example:

a.       On the other hand, the representation of single-parent families is presented as an alternative that would never be chosen (referee’s note:  this claim needs reference as it is a strong claim), otherwise it would be a cause of a problem that occurred with the couple, such as separation or divorce (vaguely written). The indications of the program lie in consoling the children and in the explanation that it continues to approach, in a certain way, homonormativity (what do you mean here?)

b.       In several of the materials (among which the materials from No. 2 to 10 and No. 14 stand out), contents framed in gender roles are expressed that can influence the perception that the families that put them in tension have of themselves to heteronormativity (What do you mean here?)

5.       Overall, the manuscript still needs to tighten its presentation of results and discussion for clarity. The paper is still replete with grammar lapses (155, 164, as examples). Consider some more editing. 

Author Response

Author's responses to the second revision

Thank you very much again for the reviews. We ask the editor when we should submit our writing for English review. I guess now is the time.

1.    There are still sections where parenthood (instead of parenting) is used.
I’m so sorry. Now we are sending the correct version. 

2.       In the abstract, the last sentence, still needs some editing: 

a.       As final reflections, we propose the need to include different perspectives that include a greater variety of families, beyond traditional representations. Especially within public policies.

We change to: It is concluded that gender stereotypes supported by heteronormative models cause the marginalization of a significant percentage of families that do not fit into the imaginary of public policies under these heteronormative models, such as LGBTIQ+ families.

b.       The investigations that have accounted for the exercise of parenthood, in this sense, in Chile, have been quite diverse. Where couples must develop a series of different strategies to become parents (Alday-Mondaca & Lay-Lisboa, 2021a; X 92 2022) where socioeconomic differences are usually determinant to achieve this, not necessarily in relation to institutionality (Herrera et al. , 2018).

We changed this to: In Chile, the research that have accounted for the exercise of parenting have been quite diverse. Some show how couples carry out actions inside and outside the institutional framework to become parents (Al-day-Mondaca & Lay-Lisboa, 2021a; X, 2022); other research has shown how socioeconomic differences are determinant in becoming parents (Herrera et al., 2018).

c.       It does not argue that the problems or experiences that a policy addresses are not real, but instead calls these conditions a represented social problem.
We changed this to: This does not mean that the problems or experiences that a policy addresses are not real, but rather that it understands these conditions to be a represented social problem.

d.       Along with the above, the images show representations that replicate stereotypes in 387 parenting in the logic of the private-internal, regarding motherhood, versus the public-388 external, where paternity is linked. Which is similar to a study that specifically analyzed the images in some materials (Maldonado, 2020).

We changed this to: Along with the above, the different forms of representation of the images present in the documents replicate differences between the constructions of maternity versus paternity. In the first case, motherhood is represented as part of the private, while fatherhood is represented as something public. This is similar to a study that specifically analyzed the images in some materials (Maldonado, 2020).

2.    I suggest that Table 1 be simplified; in fact, its inclusion may be unnecessary.  
We changed this to: Thanks for the suggestion on deleting the table. However, we consider it important to show the type of document, its name and location in case someone else wants to review it.

3.    Some lines are still written vaguely, for example:
Thank you, we tried to rewrite this section to make it more clear.

a.    On the other hand, the representation of single-parent families is presented as an alternative that would never be chosen (referee’s note:  this claim needs reference as it is a strong claim), otherwise it would be a cause of a problem that occurred with the couple, such as separation or divorce (vaguely written). The indications of the program lie in consoling the children and in the explanation that it continues to approach, in a certain way, homonormativity (what do you mean here?)

We changed this to: On the other hand, by focusing on the representation of the heterosexual couple as the only desired family model, the material analyzed represents single-parent families, not as an alternative to choose independently. Rather, the material would indicate the construction of the single-parent family as the consequence of a situation of breakup of the couple. The material indicates that children should be comforted and welcomed due to the possible problems that the separation brings.

b.    In several of the materials (among which the materials from No. 2 to 10 and No. 14 stand out), contents framed in gender roles are expressed that can influence the perception that the families that put them in tension have of themselves to heteronormativity (What do you mean here?)

We changed this to: In several of the materials, heteronormative patterns are reinforced (among which the materials from No. 2 to 10 and No. 14 stand out). These materials express content framed in gender roles that do not consider LGBTIQ+ families.

5.       Overall, the manuscript still needs to tighten its presentation of results and discussion for clarity. The paper is still replete with grammar lapses (155, 164, as examples). Consider some more editing. 

We will send the writing to an English grammar correction service.
